# Assessing Different Chronic Wasting Disease Training Aids for Use with Detection Dogs

**DOI:** 10.3390/ani14020300

**Published:** 2024-01-18

**Authors:** Amritha Mallikarjun, Ila Charendoff, Madison B. Moore, Clara Wilson, Elizabeth Nguyen, Abigail J. Hendrzak, Jean Poulson, Michelle Gibison, Cynthia M. Otto

**Affiliations:** 1Penn Vet Working Dog Center, School of Veterinary Medicine, University of Pennsylvania, Philadelphia, PA 19146, USAmadmoo@upenn.edu (M.B.M.); clarawil@upenn.edu (C.W.); etnguyen00@gmail.com (E.N.); ahendrzak@comcast.net (A.J.H.); jpp88@drexel.edu (J.P.); cmotto@vet.upenn.edu (C.M.O.); 2Wildlife Futures Program, New Bolton Center, School of Veterinary Medicine, University of Pennsylvania, Kennett Square, PA 19348, USA; mlucey@vet.upenn.edu; 3Department of Clinical Sciences and Advanced Medicine, School of Veterinary Medicine, University of Pennsylvania, PA 19104, USA

**Keywords:** chronic wasting disease, volatile organic compounds, detection dogs

## Abstract

**Simple Summary:**

Chronic wasting disease (CWD) is a fatal prion illness that affects deer, elk, moose, and other animals in the cervid family. One way to identify areas contaminated by CWD is to use detection dog–handler teams trained to find fecal matter from CWD-positive cervids. In this case, dogs are trained on white-tailed deer fecal matter. However, fecal matter from CWD-positive deer contains the infectious proteins (prions) that can spread CWD. Using this fecal matter as a training aid for these dogs can be risky, as it can contaminate soil in the training area and potentially infect other deer that come into contact with the prions. One solution is to use training aids that can replicate the odor of fecal matter from CWD-positive and CWD-negative deer but that do not contain prions; these would be safe to use in the environment. This study examined different materials, incubation times, and incubation temperatures for training aids to identify the aid that both maximizes the dogs’ detection performance and is practical to use. Based on these criteria, the best-performing aid was made of cotton and was incubated with fecal matter from CWD-positive deer for 24 h at 21 °C.

**Abstract:**

Chronic wasting disease (CWD) is a highly infectious, fatal prion disease that affects cervid species. One promising method for CWD surveillance is the use of detection dog–handler teams wherein dogs are trained on the volatile organic compound signature of CWD fecal matter. However, using fecal matter from CWD-positive deer poses a biohazard risk; CWD prions can bind to soil particles and remain infectious in contaminated areas for extended periods of time, and it is very difficult to decontaminate the affected areas. One solution is to use noninfectious training aids that can replicate the odor of fecal matter from CWD-positive and CWD-negative deer and are safe to use in the environment. Trained CWD detection dogs’ sensitivity and specificity for different training aid materials (cotton, GetXent tubes, and polydimethylsiloxane, or PDMS) incubated with fecal matter from CWD-positive and CWD-negative deer at two different temperatures (21 °C and 37 °C) for three different lengths of time (6 h, 24 h, and 48 h) were evaluated. Cotton incubated at 21 °C for 24 h was identified as the best aid for CWD based on the dogs’ performance and practical needs for training aid creation. Implications for CWD detection training and for training aid selection in general are discussed.

## 1. Introduction

Chronic Wasting Disease (CWD) is a highly infectious prion-related transmissible spongiform encephalopathy (TSE) that causes fatal neurodegeneration in cervid species, such as white-tailed deer, mule deer, elk, and moose [1]. CWD is further characterized by highly variable pre-symptomatic incubation periods, which are estimated to have an average duration range of approximately 2 to 4 years [2]. Infected pre-symptomatic and symptomatic individuals shed CWD prions through biological excretions, such as fecal matter, saliva, blood, and urine, which introduce the disease to the inhabited environment [3]. Currently, CWD is always fatal, and there are no vaccines or therapies available. CWD occurs at a low prevalence in most areas, with initial focal areas of infection near the point of introduction [4,5,6]. This highlights the importance of early detection methods for curbing the spread of CWD.

A promising method for controlling the spread of CWD is the use of detection dogs. These dogs can be trained to identify CWD-infected animals via their excretions, facilitating the tracking and identification of infected cervids. CWD does not appear to be a risk to dogs; dogs have been found to be generally resistant to prion diseases [7,8]. Additionally, studies on coyotes, which are phylogenetically close to dogs, have shown that coyotes are unaffected by consuming meat from CWD-positive cervids [9]. As such, dogs can safely detect CWD with little likelihood that they will contract the disease itself. Using dogs to perform premortem, noninvasive monitoring is particularly relevant, as many cervids do not show symptoms typical of CWD for months to years, making detection via visually tracking the animals unfeasible. Dogs have been, and continue to be, successfully used in a variety of conservation and disease detection efforts [10,11]. Recent studies demonstrate that detection dogs can recognize fecal samples from CWD-positive deer both in laboratory and applied settings [12].

Training on fecal matter from deer infected with CWD poses environmental risks; fecal matter from CWD-positive deer contains CWD prions, and these prions show resistance to natural deactivation mechanisms such as desiccation, degradation via freeze–thaw cycles, extracellular enzymes from bacteria and fungi, UV irradiation, or digestion by macrofauna in soil [13,14]. Additionally, handling fecal samples from CWD-positive deer has been suggested to have a high likelihood of facilitating environment recontamination [15,16]. As such, training detection dogs with fecal samples from CWD-positive deer while monitoring containment and proper handling may be unfeasible for canine handlers. Because of potential procedural error and CWD’s characteristic resistance to deactivation, alternative methods for training CWD detection dogs require further investigation.

One possible solution to this biocontamination concern is the use of training aids, which are substances that are not the target substance but either mimic or ad/absorb the target odor. A training aid would be greatly useful if it were able to retain a distinguishable odor of samples from CWD-positive deer, as it would not contain the infectious prions and risk contamination of additional environments. For certain substances, like narcotics and explosives, there are pseudo-odors that smell like the target odor that can be used for training. The use of pseudo-odors requires a clear understanding of the chemical components of the target odor and how the odor may change as it is exposed to different types of conditions [17]. Given the complexity of biological odors, a better option for a safe training aid would be to use materials that can ab/adsorb the odor of a target substance [17]. Ab/adsorptive training aids come in many forms and are composed of various substrate types, ranging from readily available items like gauze and cotton balls to specialized materials like polydimethylsiloxane (PDMS), an inert silicone polymer [18], and commercially made items like the polymer-based GetXent tubes [19]. Cotton/fiber and polymer-based materials ad/absorb odor from a target material without direct contact. These aids are kept near the intended true material and absorb VOCs over time. 

Absorptive or adsorptive training aids have been used to train dogs to detect different categories of biological odors, especially in cases where the target substance is infectious or difficult to procure. For example, dogs learned to detect Bovine Diarrheal Virus, which is highly contagious among cattle, using a polymer-based training aid incubated with cultured material from infected animals [20]. Human remains detection dogs can be trained using absorptive training aids made from cotton [21]. Studies on COVID-19 detection have used the Getxent tube [19] to safely collect samples from infected patients [22]. While these studies suggest that training aids could be useful for biological odor detection, there are no current studies on biological odor detection that compare the efficacies of different types of training aids. 

Incubating training aids with fecal samples from CWD-positive deer could be valuable, as infected fecal samples are difficult to access and the handling of such material poses a risk. However, the utility of different training aids depends on several factors, such as the odor profile of the target substance, the temperature during incubation, and the duration of incubation with the target odor [18,23]. Given the risk of contamination of the environment and the growing interest in using detection dogs operationally to detect CWD, it is imperative to explore which training aid type, and preparation method, is most effective for training CWD detection dogs.

The extent to which dogs can classify samples as coming from CWD-positive or CWD-negative deer could be considered a form of learning described as “concept formation”, wherein an animal learns to identify the complex and variable patterns that are indicative of a disease [24,25]. Whether a detection dog classifies a novel sample (not previously presented to them in training) as positive or negative can be impacted by a variety of factors, including, but not limited to, their reinforcement history [26], inherent bias [27], the type of final response (e.g., sit, stand-and-stare, etc.) [28], and the required duration of the trained final response [29]. However, if a novel odor is markedly distinct from a dog’s trained concept of a CWD-positive odor, they will be unlikely to recognize this novel odor as their target.

This study examined (1) the extent to which dogs can discriminate between samples from CWD-positive and CWD-negative deer in non-fecal training aids and (2) the effect of training aid type (cotton ball, PDMS, and GetXent), incubation time, and incubation temperature on trained dogs’ ability to discriminate between samples from CWD-positive and CWD-negative white-tailed deer.

## 2. Materials and Methods

### 2.1. Participants

Following institutional animal care and use committee approval (protocols 806895 and 806913), 4 privately owned dogs (2 Labrador retrievers, 1 Finnish Spitz, and 1 Pembroke Welsh Corgi) and 2 dogs owned by the Penn Vet Working Dog Center (2 German Shepherds) were enrolled (Table 1). Handlers of the privately owned dogs were recruited from the Penn Vet Working Dog Center community science program. The handlers signed informed consent to allow their dog to be trained to detect CWD in the fecal samples of white-tailed deer (*Odocoileus virginianus*). 

### 2.2. Materials

#### 2.2.1. Samples

Sample procurement: Samples were collected in the same manner as that described in the study by Mallikarjun et al. [12]. Fecal samples from CWD-positive and CWD-negative white-tailed deer were procured from the United States Department of Agriculture. The University of Pennsylvania’s School of Veterinary Medicine’s Wildlife Futures Program (WFP) procured, aliquoted, and distributed all of the fecal samples from CWD-positive and CWD-negative deer. Samples were maintained at −20 °C or colder until use by the Penn Vet Working Dog Center. Deer were tested via IHC in either the obex (brain) or RPLN. Fecal samples were collected in four US regions defined as West, Midwest, South, and East. Testing for CWD was performed at the National Veterinary Services Laboratory.

These samples were shipped in 4 oz glass Mason jars from WFP to the Penn Vet Working Dog Center. Samples ranged from 0.5 g to 2 g in weight. New samples were periodically shipped from WFP and incorporated into sessions over the course of the study.

All samples had a unique identification number used to randomize the dogs’ encounters with positive and negative samples.

#### 2.2.2. Fecal Matter Used to Create Training Aids

To prevent confounds due to the representativeness of individual positive and negative WTD fecal samples, pooled positive and negative samples were used. Pooled samples were used to ensure that dogs’ performance with respect to any given alternative aid was not dependent on the sample itself. Samples differed widely based on deer age, sex, location (which influences diet), and disease progression. Additionally, use of a single positive or negative across all aids would not have been possible, as there would not have been enough fecal material for eighteen different incubations, and incubations should not be performed more than once with any given sample.

Prior to the testing phase, the WFP pooled 20 positive samples and 20 negative samples from the set of collected and tested samples (see sample procurement above for more information on the sample collection and identification of CWD status). This generated a positive blended mixture and a negative blended mixture, respectively. The positives were selected by examining canine performance with respect to known positive samples used to train dogs, as presented in Mallikarjun et al. [12], and selecting 20 samples for which all 3 original CWD detection dogs successfully executed their trained final response behavior. Similarly, 20 known negative samples for which all 3 original CWD detection dogs successfully passed were chosen. The pooled positive and pooled negative samples were aliquoted by the WFP into 4 oz jars containing 2 g of pooled fecal matter each.

The dogs were trained on the pooled aliquots for 4–5 sessions to ensure that they were able to identify and perform their trained final response behavior for the pooled positive sample and ignored the pooled negative sample.

#### 2.2.3. Training Aids

WFP created training aids by incubating different materials with the pooled fecal samples from either positive or negative white-tailed deer. Three aid materials were used: cotton balls, GetXent tubes [19], and PDMS, a polymer that holds odors [18]. The GetXent tubes were donated to the PVWDC by GetXent (Neuchâtel, Switzerland).

The three different materials were each incubated for one of three lengths of time (6, 24, or 48 h) at 2 different temperatures (21 °C and 37 °C) for a total of 18 different types of aids (see Table 2). Each of the 18 types of aid was incubated with positive pooled fecal matter, negative pooled fecal matter, or nothing (as a control).

### 2.3. Procedure

#### 2.3.1. Participant Training

The six dogs had previously participated in scent detection studies at the Penn Vet Working Dog Center. All dogs were originally trained to detect a synthetic chemical training odor (Universal Detector Calibrant, or UDC) [30] on an eight-armed scent detection wheel. Dogs were trained to search the wheel in order. They executed a trained final response at the port containing the UDC odor (either a 2 s stand-and-stare procedure or a sit), as well as a ‘blank wheel’ behavior, which consisted of leaving the search area to sit on a table to indicate that the target odor was not present in the presented odor samples. The dogs’ correct behavior was marked with a conditioned secondary reinforcer (“clicker”), and dogs were rewarded with a primary reinforcer (food) for finding and correctly performing their trained final response behavior at the port containing the UDC odor. During these training sessions, the dogs were trained to ignore laboratory odors, such as nitrile gloves, isopropyl alcohol, and paper towels. Unimpregnated versions of the aids (cotton balls, GetXent tubes, and PDMS) were also used as distractors for these dogs in training.

Three of the privately owned dogs had previously been trained to find fecal samples from CWD-positive deer in laboratory and field settings. These additional three dogs were trained to differentiate fecal samples from CWD-positive and CWD-negative deer specifically for this alternative training aids study. As above, unimpregnated cotton balls, GetXent tubes, and PDMS were used as distractors during training, so dogs were familiar with the odor of the unimpregnated aid. All dogs achieved at least 80% sensitivity and 80% specificity with respect to their initial response to odor across three practice sessions prior to participating in this study. The previously trained dogs required between 5 and 8 sessions of 8–10 trials each to satisfy the criteria. The three dogs that were trained specifically for this study required 24 sessions of 8–10 trials each to reach 80% sensitivity and 80% specificity across three sessions. Appendix A provides training and testing phase information in a table.

#### 2.3.2. Equipment and Setup

For each trial, samples were set up in a five-port lineup (Figure 1). The five-port line-up was constructed in a small, low-distraction room with which all the dogs were familiar. This lineup was used rather than the scent detection wheel because a subset of the dogs in this study performed searches better in this manner, and the search format did not make a difference for the other dogs. Ports contained training aids (CWD-positive, CWD-negative, or control) or distractor items consisting of laboratory odors (e.g., nitrile gloves, isopropyl alcohol, and paper towel).

A Panasonic HC-V380 video camera (Panasonic, Philadelphia, PA, USA) was used to record the experimental sessions.

#### 2.3.3. Testing

Dogs participated in four double-blind tests. No fecal matter was presented in these tests; dogs only saw novel aids. Across these four tests, dogs were presented with each of the three types of aids (cotton ball, GetXent, and PDMS) incubated at two temperatures (21 °C and 37 °C) for three lengths of time (6, 24, or 48 h). Trials either contained a positive sample (hot trials) or did not contain a positive sample (blank trials). Each hot trial contained the CWD-positive, CWD-negative, and control versions of each aid; for example, one hot trial contained CWD-positive, CWD-negative, and control cotton balls incubated at 21 °C for 6 h, as well as two distractors. As such, there were 18 total hot trials across all tests—one for each type of aid. There were 10 total blank trials across all tests. Blank trials contained only control aids and distractor items.

In each test trial, the order of the CWD-positive, CWD-negative, and control samples was randomized. Additionally, across tests, the trials were randomized such that no dogs saw the alternative training aids in the same order. Each session included 2–3 blank trials and 4–5 positive sample trials. Appendix A provides training and testing phase information in a table.

In each trial containing a CWD-positive training aid, dogs were marked as correct if they produced an alert for the CWD-positive training aid. Dogs were also marked as correct for a blank trial if they sniffed and passed every item in the lineup and performed their blank wheel behavior (sitting on a table at the end of the lineup). If a dog sniffed and passed the CWD-positive aid in a hot trial, once the dog finished the trial, the handlers provided a neutral verbal marker (e.g., “ok!”) and a low-value reward (e.g., kibble) and reset the dog for the next trial.

Four dogs were trained to produce an alert regarding target samples on their first encounter with the target sample. The fifth dog, Charlie, developed several mechanical difficulties with searching during the testing phase and was eventually allowed to free-search the samples (i.e., it was not necessary to search the samples in order). Charlie’s data were kept separate from the other four dogs.

### 2.4. Analyses

Statistical analyses were carried out in R version 4.2.0.

#### 2.4.1. Video Coding

Videos were recorded for each testing session in this study. The videos were filmed at 60 frames per second. These data were coded using BORIS video coding software [31] to note the duration the dogs spent sniffing each odor in the lineup. Duration at odor source was defined for coders as the amount of time dogs’ noses were within 6 inches of the port holding the odor. One coder coded all the test videos, and a second coder coded a subset of the test videos for inter-rater reliability. When the IRR was determined to be acceptable between the main coder and the second coder, the main coder’s data were used for the statistical analyses. Appendix A contains the ethogram used for coding.

Using R, an inter-rater analysis was conducted to determine the coders’ reliability in assessing duration at odor source for 5 randomly chosen videos out of the 25 total test videos. The lengths of time spent at the ports were compared between coders. The average duration difference between the two coders was 0.069 s. The inter-rater analysis was conducted based on a single-rating, consistency, one-way mixed-effects model. This model showed that the intraclass coefficient was 0.967, with a confidence interval ranging from 0.956 to 0.975. This is considered excellent reliability [32].

Model 1: *How do Sample Type, Temperature, and Time affect the occurrence of the dogs’ trained final response with different training aids, and how do these factors interact with CWD Status?*

This analysis only included the four dogs who searched linearly. 

A binomial linear mixed-effects model fit was attempted, with Sample Type (cotton, PDMS, and GetXent), Incubation Temperature (21 °C and 37 °C), Incubation Time (6, 24, and 48 h), and CWD Status (positive or negative) serving as fixed effects and Dog serving as a random effect. This model did not converge. As a result, a reduced binomial linear mixed-effects model was used to examine the individual 2-way interactions between CWD Status and the training aid features (Sample Type, Incubation Temperature, and Incubation Time). The random effect was Dog identity. This model was analyzed via Type II Wald chi-square test.

#### 2.4.2. Change in Behavior Analyses

While it is expected that dogs will perform their trained final alert on what they consider target samples, Dechant proposed that dogs may not execute their full trained final response if there are any changes in the search context from the original training scenario or other disruptions of the search conditions [33]. In this case, dogs were trained on fecal matter from CWD-positive deer, but they saw training aids incubated with CWD-positive fecal matter during the test. Previous studies have suggested that changes in the target medium can reduce the rate of trained final alerts for the target odor; however, dogs still show a change in behavior for the new target medium, suggesting that they still recognize the odor in a different medium and can differentiate it from the control and negative samples [34].

As such, a second analysis was conducted here with a less stringent criterion: change in behavior at the sample. This was measured via dogs’ time spent at the source of the odor. Given that dogs spent, on average, 0.65 s (st. dev. = 0.30 s) checking control odors in previous studies [12,34], if the dogs spent 1 s or more at any odor source, this activity was considered a change in behavior. 

Model 2: *How do Sample Type, Temperature, and Time affect change in behavior for different training aids, and how do these factors interact with CWD Status?*

This analysis only included the four dogs who searched linearly.

As in Model 1, a binomial linear mixed-effects model fit was attempted, with Sample Type, Incubation Temperature, Incubation Time, and CWD Status serving as fixed effects and Dog serving as a random effect. This model did not converge. As a result, we again used reduced binomial linear mixed-effects models examining the individual interactions between CWD Status and the training aid features (Sample Type, Incubation Temperature, and Incubation Time) and analyzed this model via Type II Wald chi-square test.

## 3. Results

### 3.1. Dogs’ Trained Final Alerts for Samples

Four out of the five dogs trained in this study searched all the samples in order, without comparing the samples in the lineup to one another. Upon these four dogs’ initial encounter with the samples, they performed their trained final alert on 64/90 positive aids (71%) and alerted incorrectly for 28/69 negative samples (41%) and 6/80 control samples (8%). In a perfect performance, the dogs would alert for every positive sample and not alert for any negative or control samples. Below, Table 3, Table 4 and Table 5 show the proportion of dogs’ alerts for the CWD-positive, CWD-negative, and control aids broken down by material, temperature, and time, respectively.

In contrast, in our previously conducted field study, the tested dogs found 8 out of 11 positive samples (73%) and incorrectly alerted for 14/78 negative samples (18%). While the dogs’ sensitivities are similar between the field study and this study, the false alert rate is much higher for the dogs using the training aids, which suggests that the training aids are not identically representing the samples—specifically the negative samples.

### 3.2. Trial Accuracy in Training Aid Detection

Four of the five dogs that participated in this study searched all the samples in order, and their initial response to the odor was used as their behavior exhibited at a sample. This was shown in the Dogs’ Trained Final Alerts for Samples section above. However, one dog that participated in this study did not search in order and instead free-searched, meaning he searched all the samples non-linearly and was allowed to go back and forth between samples and compare them to one another before alerting. This dog could not provide an accurate initial response to individual samples since he often checked samples multiple times before providing a response, but he could provide trial accuracy data. 

Examining dogs’ performance according to trial accuracy allows for the inclusion of all the dogs that participated in this study, and this can provide additional information about the best-performing aid. The result of a trial was considered correct if the dog performed a trained final alert for the positive sample and did not alert for the negative and/or control samples. For Charlie, who free-searched, correct trials consisted of those in which he consistently passed the negative and control aids (even if he smelled them multiple times) and issued an alert for the positive aid. Trial accuracies according to material, temperature, and time are shown in Table 6, Table 7 and Table 8.

Model 1: *How do Sample Type, Temperature, and Time affect the occurrence of the dogs’ trained final response with different training aids, and how do these factors interact with CWD Status?*

There were no significant interactions between CWD Status and Sample Type, CWD Status and Temperature, or CWD Status and Time. There were no significant main effects of Sample Type (X^2^ = 0.254, *p* = 0.89), Temperature (X^2^ = 0.10, *p* = 0.75), and Time (X^2^ = 0.12, *p* = 0.73). There was a significant main effect of CWD Status (X^2^ = 58.76, *p* < 0.0001) such that dogs performed their trained final response more often for CWD-positive samples than CWD-negative or control samples (CWD-positive versus CWD-negative: z = 3.73, *p* = 0.0002; CWD-positive versus control: z = 7.99, *p* < 0.0001). This confirms the above finding that dogs can detect CWD odor on the training aids. The dogs also performed their trained final response more often for the CWD-negative samples (an incorrect trained final response) than the control samples (z = −4.94, *p* < 0.0001), potentially because the fecal matter odor was still strong on the CWD-negative samples. 

These findings demonstrate that the dogs more commonly performed their trained final response behavior for CWD-positive aids than other non-target aids, but there was no difference in the number of trained final responses to the different types of training aids (cotton ball, PDMS, and GetXent) or to the aids incubated at different temperatures (21 °C and 37 °C) and for different amounts of time (6, 24, and 48 h) (see Figure 2). 

Model 2: *How do Sample Type, Time, and Temperature affect the occurrence of dogs’ change in behavior for different training aids, and how do these factors interact with CWD Status?*

There was a significant main effect of CWD Status (X^2^ = 64.86, *p* < 0.0001) such that dogs showed changes in behavior more often for the CWD-positive samples than the CWD-negative or control samples (CWD-positive versus CWD-negative: z = 3.06, *p* = 0.002; CWD-positive versus control: z = 98.54, *p* < 0.0001). The dogs also showed changes in behavior more often for the CWD-negative sample than the control (z = −5.69, *p* < 0.0001). There were no main effects of Temperature (X^2^ = 0.20, *p* = 0.65), Time (X^2^ = 0.01, *p* = 0.98), or Material (X^2^ = 4.42, *p*= 0.11).

There were no significant interactions between CWD Status and Material (X^2^ = 7.84, *p* = 0.10) or CWD Status and Time (X^2^ = 0.03, *p* = 0.99). There was a significant interaction between CWD Status and Temperature (X^2^ = 6.055, *p* = 0.048). Post hoc tests with a Tukey adjustment showed that the dogs demonstrated significantly more changes in behavior for the 21 °C CWD-positive samples than the 21 °C CWD-negative samples (z = 2.81, *p* = 0.014), while there was no difference in the frequency of change in behavior for the samples between the 37 °C CWD-positive samples and the 37 °C CWD-negative samples (z = 1.62, *p* = 0.24). See Figure 3 for results.

## 4. Discussion

This study aimed to (1) assess whether trained CWD detection dogs could identify the signature odor of CWD in training aids and (2) examine the extent to which training aid material, incubation temperature, and incubation time affected the aids’ representations of the signature odor of CWD as determined by detection dogs. Dogs previously trained to discriminate between fecal matter from CWD-positive and CWD-negative deer were tested on different types of training aids in a double-blind scenario. The dogs successfully discriminated between CWD-positive, CWD-negative, and control training aids based on their trained final responses and changes in behavior at the samples. The dogs showed changes in behavior significantly more frequently at CWD-positive aids than CWD-negative aids when the aids were incubated at 21 °C but not when the aids were incubated at 37 °C.

Dogs trained on CWD fecal matter performed their trained final response and showed changes in behavior significantly more frequently for the CWD-positive training aids than the CWD-negative or control training aids. This suggests that the CWD-positive and CWD-negative training aids used in this study reflect, to some degree, the difference between the CWD-positive odor and the CWD-negative odor in fecal matter. Biological odors are complex; one VOC analysis of CWD-positive and CWD-negative fecal matter found 182 different VOCs that differed between CWD-positive and CWD-negative fecal matter and, after eliminating VOCs associated with diet and those not associated with selection criteria, found seven different characteristic VOCs. In general, there are several different VOCs that are related to any disease signature. The rates at which molecules adsorb onto or are absorbed into different mediums can differ depending on the aid composition, temperature, and the time of incubation, which could disrupt the proportions of each molecule in the aid. Here, for three different aid materials at different incubation temperatures and times, the dogs alerted more frequently issued alerts for the CWD-positive samples (trained final alerts on 71% of the samples) than the CWD-negative samples (trained final alerts on 41% of the samples). However, even though the dogs issued alerts more frequently for the positive samples than the negative samples, a 41% false alert rate is relatively high, and it is higher than the dogs’ false alert rate for fecal matter from CWD-negative deer. The dogs’ successful differentiation suggests that ad- or absorptive training aids can be used at least adjunctively alongside CWD fecal matter for training CWD detection; however, actual fecal matter must also be used in training for the dogs to be successful in the field since the aids do not perfectly replicate the odor.

The dogs did not exhibit any differences in their trained final response behavior based on training aid material, temperature, or time. Dogs may perform their trained final response at a lower rate if there are any disruptions to the original trained search context [33]; in this case, the target odor was presented in novel media not present in the training sessions (training aids versus fecal matter). Additionally, the proportions of odor may have been slightly different than those expected from fecal matter, which could also reduce the dogs’ rate of response. The use of aids may have caused the dogs to display reduced-criteria changes in behavior for a target aid, like hesitating at a sample for between 1 and 2 s rather than performing a full 2-or-more-second trained final alert. As such, when using training aids with dogs that have been trained on the actual target substance, handlers should be prepared for the eventuality that their dogs may exhibit changes in behavior around the target odor in a training aid as compared to their full trained final response.

Incubation temperature affected dogs’ changes in behavior at the CWD-positive and CWD-negative samples; the dogs showed more changes in behavior at positive samples than negative samples incubated at 21 °C, while there was no difference in the dogs’ behavior for positive and negative samples incubated at 37 °C. One potential reason for this difference may be that keeping samples at a higher temperature can result in changes in the microbiome [35], which, as found in studies on the fecal matter of other species, can alter VOCs. Warmer temperatures can change the ratio of bacterial species in fecal matter [36]. Given that bacterial metabolites likely contribute to the VOCs produced by fecal matter [37,38], it is necessary to keep the bacteria levels as true to the original sample as possible when incubating training aids. As such, 21 °C is a better incubation temperature to ensure that the VOC signature in the training aids is as close to the target odor as possible.

While there was no effect of incubation time on the dogs’ trained final alerts or changes in behavior, the length of time samples spend in warmer temperatures, which can affect the bacteria in a sample, must be considered. Incubation times for training aids vary widely across the field depending on aid material, target odor, incubation temperature, and whether incubation was carried out passively or dynamically, that is, via pulling air over or through the aid [17,39]. In general, because heating increases vapor pressure and increases VOC availability [18], incubation times reduced as incubation temperature rose. There is not a clear choice of incubation time for CWD aids given the current data; further data are needed to determine if there is an ideal incubation temperature for CWD training aids.

Given that the aid materials elicited relatively similar performance in both dogs’ trained final alerts and changes in behavior, factors such as cost and ease of use can come into consideration when selecting a training aid. The GetXent tubes, which weigh 0.93 g each, currently cost USD 10 for 1 tube, or a bag of 50 tubes can be purchased for USD 269, which averages to about USD 5.38 per tube [19]. The cost for PDMS varies but is approximately USD 0.07 to 0.21 per gram [40,41]. Cotton balls are the cheapest option at USD 0.01 or lower per cotton ball [42]. When considering ease of use, PDMS requires a precise mixing and baking process prior to its incubation with the target odor. While the instructions are straightforward, the aids have to be prepared ahead of time, and there are many opportunities in the process for accidental contamination. GetXent tubes and cotton balls, in contrast, are immediately ready for incubation. Another factor regarding ease of use is the amount of time the aids retain odor for repeated use; further studies should explore different storage conditions and the duration of time the aids continue to hold odor. Additionally, from observation of our own aids, it is clear that cotton can easily pick up dirt from the environment, while the Getxent tubes and PDMS are more resistant. Studies should explore how well the aids retain their function when used outdoors or in environments that could potentially contaminate the aids.

One potential limitation of our work is that we were unable to assess deer for diseases other than CWD when performing sample collection. We aimed to collect a wide variety of white-tailed deer fecal samples from across the country [12]. The samples were diverse in terms of age, sex, and location, but there was no way to determine if the deer who did not have CWD had other illnesses. When conducting biological detection studies related to diseases, it is a good practice to include different disease controls as negative samples to ensure that dogs are homing in on the target disease odor and not on odors characteristic of more general sickness or inflammation [23]. Given that we collected hundreds of samples of wild deer, it is likely that some of these CWD-negative deer were not completely healthy, but there was no way to be certain. Showing the dogs a wide variety of samples, as we did in initial training, is the best way to account for this potential limitation.

This study examined the performance of training aids in a laboratory setting, specifically focusing on the closeness of the training aid odor to the target CWD fecal matter odor. In a field environment, there are several other potential considerations for training aids alongside the accurate emulation of the target odor. If the aid is a single-use aid, meaning that it is hidden once in the environment for the dog to find and then discarded, it must be inexpensive and easy to make, without impacting the quality of the stored fecal matter, which is required not just for training but for other CWD research. If the aid is meant to be used multiple times, the aid must not pick up extraneous environmental odors and must be somewhat resistant to physical contamination such as with dirt or water. Further studies are needed to explore how these aids function in a field environment.

## 5. Conclusions

This study compared the utility of three different training aids for detecting CWD (cotton, GetXent tubes, and PDMS) incubated at two different temperatures (21 °C and 37 °C) for three different lengths of time (6 h, 24 h, and 48 h). Detection dogs trained to issue alerts upon detecting CWD-positive fecal matter performed their trained final response significantly more often for the CWD-positive training aids, regardless of temperature, time, and material. When examining the dogs’ change in behavior at the samples, there was an effect of temperature in which the dogs differentiated positive and negative samples incubated at 21 °C but did not do so for samples incubated at 37 °C. Further research is needed to assess the strengths and weaknesses of the different training aids in a functional field environment.

## Figures and Tables

**Figure 1 animals-14-00300-f001:**
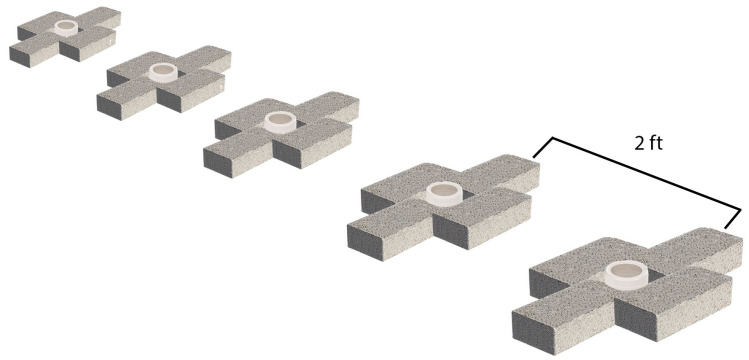
A diagram of the setup of the bricks and sample jars for training.

**Figure 2 animals-14-00300-f002:**
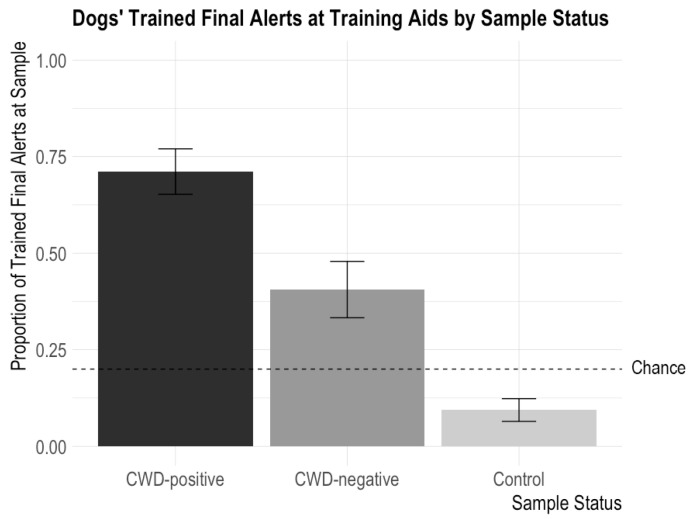
Dogs’ proportions of trained final responses to training aids according to CWD status (CWD-positive, CWD-negative, and controls). The error bars show standard error. Dogs issued alerts significantly more often for CWD-positive aids than CWD-negative aids and significantly more often for both CWD-positive and CWD-negative aids than control samples. The highest possible chance level was 0.20; chance could be lower than 0.2 based on the number of “blank” trials in the session.

**Figure 3 animals-14-00300-f003:**
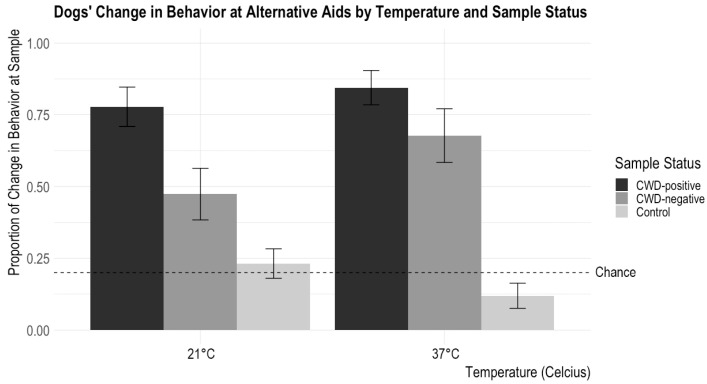
Dogs’ proportions of changes in behavior at a sample according to incubation temperature and sample status are shown. The error bars show standard error. Chance was set at about 20% (it could be slightly lower depending on the number of blank trials—trials without a target odor—in a session). Dogs showed changes in behavior for CWD-positive samples significantly above the chance level, and, overall, they showed changes in behavior at CWD-positive samples more than the CWD-negative aids. Dogs showed significantly more changes in behavior at the CWD-positive sample incubated at 21 °C than the CWD-negative sample incubated at 21 °C. In contrast, there is no difference in change in the behavior proportion between CWD-positive and CWD-negative samples incubated at 37 °C.

**Table 1 animals-14-00300-t001:** This table contains information about the dogs’ sex, age, and breed and the odors that they learned to recognize prior to participating in this study. All of these dogs were trained at the Penn Vet Working Dog Center.

Dog Name	Sex	Age (Years)	Breed	Learned Odors Prior to This Study
Jari	M	8	Finnish Spitz	CWD, UDC ^1^
Kiwi	F	8	Labrador Retriever	CWD, UDC ^1^
Charlie	M	8	Labrador Retriever	CWD, UDC ^1^
Crunch	M	4	Pembroke Welsh Corgi	Sinonasal Inverted Papilloma, UDC ^1^
Bobbie	F	5	German Shepherd	Ovarian Cancer, UDC ^1^
Osa	F	9	German Shepherd	Ovarian Cancer, UDC ^1^

^1^ Universal detector calibrant.

**Table 2 animals-14-00300-t002:** This table shows all 18 types of training aids used in the study. Materials included cotton, PDMS (polydimethylsiloxane), and GetXent tubes. There were two potential temperatures of incubation (21 °C and 37 °C) as well as 3 potential incubation times (6, 24, or 48 h) for a total of 18 different types.

Training Aid Material	Temperature	Time
Cotton ball	21 °C	6 h
Cotton ball	21 °C	24 h
Cotton ball	21 °C	48 h
Cotton ball	37 °C	6 h
Cotton ball	37 °C	24 h
Cotton ball	37 °C	48 h
PDMS	21 °C	6 h
PDMS	21 °C	24 h
PDMS	21 °C	48 h
PDMS	37 °C	6 h
PDMS	37 °C	24 h
PDMS	37 °C	48 h
GetXent tube	21 °C	6 h
GetXent tube	21 °C	24 h
GetXent tube	21 °C	48 h
GetXent tube	37 °C	6 h
GetXent tube	37 °C	24 h
GetXent tube	37 °C	48 h

**Table 3 animals-14-00300-t003:** Trained Final Alerts (TFAs) for chronic wasting disease (CWD)-positive, CWD-negative, and control samples according to sample material. Trained final alerts are shown as percentages and as number of trained final alerts over total number of samples seen by the dogs in a given category.

Incubation Material	CWD-Positive	CWD-Negative	Control
Cotton	73.3% 22/30	38.1% 8/21	11.5% 3/26
GetXent tube	73.3% 22/30	36.3% 9/22	3.7% 1/27
PDMS *	66.7% 20/30	42.3% 11/26	7.4% 2/27

* Polydimethylsiloxane.

**Table 4 animals-14-00300-t004:** Trained Final Alerts (TFAs) for CWD-positive, CWD-negative, and control samples according to sample incubation temperature. Trained final alerts are shown as percentages and as number of trained final alerts over total number of samples seen by the dogs in in a given category.

Incubation Temperature	CWD-Positive	CWD-Negative	Control
21 °C	44.4% 20/45	34.2% 13/38	11.3% 5/44
37 °C	73.3% 33/45	48.3% 15/31	2.7% 1/36

**Table 5 animals-14-00300-t005:** Trained Final Alerts (TFAs) for chronic wasting disease (CWD)-positive, CWD-negative, and control samples according to sample incubation time. Trained final alerts are shown as percentages and as number of trained final alerts over total number of samples seen by the dogs in a given category.

Incubation Time	CWD-Positive	CWD-Negative	Control
6 h	66.7% 20/30	36% 9/25	7.7% 2/26
24 h	73.3% 22/30	56% 14/25	6.6% 2/30
48 h	73.3% 22/30	26% 5/19	8.3% 2/24

**Table 6 animals-14-00300-t006:** Chronic wasting disease detection trial accuracy according to material. Trial accuracy is shown as number of correct trials divided by the number of total trials for each of the incubation materials.

Incubation Material	Correct Trials
Cotton	24/36
GetXent tubes	14/36
PDMS *	18/36

* Polydimethylsiloxane.

**Table 7 animals-14-00300-t007:** Chronic wasting disease detection trial accuracy according to temperature. Trial accuracy is shown as the number of correct trials divided by the number of total trials for each of the incubation temperatures.

Incubation Temperature	Correct Trials
21 °C	25/54
37 °C	31/54

**Table 8 animals-14-00300-t008:** Chronic wasting disease detection trial accuracy according to time. Trial accuracy is shown as the number of correct trials divided by the number of total trials for each of the incubation times.

Incubation Time	Correct Trials
6 h	21/36
24 h	14/36
48 h	21/36

## Data Availability

The data presented in this study are available on request from the corresponding author. The data are not publicly available due to current use in other studies.

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
