# Peer review of "Assessing Different Chronic Wasting Disease Training Aids for Use with Detection Dogs"

_animals, 2024, doi:10.3390/ani14020300_

Round 1

Reviewer 1 Report

Comments and Suggestions for Authors

This paper assessing the odor soaks as training aids for detection of animals with CWD by canine was very well-written and easy to read and follow. I found that the experimental and data analysis methods were sound, the discussion thorough, and the conclusions were appropriately supported by the results. I believe that this manuscript only needs minor revisions as follows:

In the introduction, I think it would be useful for the reader to have more information about the types of training aids used for detection of other types of disease, as well as a brief review comparing research that has been done with odor soaks.

The methods section is very well-written; however, there are some inconsistencies with the paragraph indentations throughout this section. 

The first paragraph of the results section mentions Tables 1, 2, and 3, but I believe this should be Tables 2-4. I am a bit confused about how the data for Tables 5-7 were calculated as compared to the data in Tables 2-4. This should be explained better. Also, be sure to reference Tables 5-7 somewhere in the text.

I enjoyed the discussion. It covers all relevant topics and summarizes the findings in a broader context very well. 

Author Response

Thank you for your comments. Our responses are below. 

This paper assessing the odor soaks as training aids for detection of animals with CWD by canine was very well-written and easy to read and follow. I found that the experimental and data analysis methods were sound, the discussion thorough, and the conclusions were appropriately supported by the results. I believe that this manuscript only needs minor revisions as follows:

In the introduction, I think it would be useful for the reader to have more information about the types of training aids used for detection of other types of disease, as well as a brief review comparing research that has been done with odor soaks.

Thank you, this change has been made in the introduction to talk more about the use of training aids in other biological detection scenarios and about, more generally, the use of ab/adsorptive training aids. 

The methods section is very well-written; however, there are some inconsistencies with the paragraph indentations throughout this section. 

We have gone through and standardized the paragraph indentations.

The first paragraph of the results section mentions Tables 1, 2, and 3, but I believe this should be Tables 2-4. I am a bit confused about how the data for Tables 5-7 were calculated as compared to the data in Tables 2-4. This should be explained better. Also, be sure to reference Tables 5-7 somewhere in the text.

I have added clarifying language in Section 3 (Results) indicating that tables 2, 3, and 4 are referring to trained final alerts on initial encounter with each sample (meaning that in a perfect performance, dogs would have a 100% for the positive samples, meaning they alerted on all of them, and a 0% for the negative and control samples, meaning they did not alert on any of these. 

In contrast, tables 5, 6, and 7 are showing trial accuracy as compared to dogs’ performance on individual samples.  This is explained in the section “trial accuracy in training aid detection”. 

Four of the five dogs that participated in this study searched all samples in order, and their initial response to odor was used as their behavior at sample. This was shown in the dogs’ trained final alerts on samples section above. However, one dog that participated in this study did not search in order and instead free-searched, meaning he searched all the samples non-linearly and could go back and forth between samples and compare them to one another before alerting. This dog cannot provide accurate initial response to individual samples, since he often checked samples multiple times before providing a response, but he can provide trial accuracy data.

I enjoyed the discussion. It covers all relevant topics and summarizes the findings in a broader context very well. 

Thank you.

Reviewer 2 Report

Comments and Suggestions for Authors

Overall I think this is an excellent and highly applied piece of research and a very well written manuscript. Congratulations to all authors for your work! I have some minor comments and suggestions below. 

-        - Simple summary and abstract both provide excellent information and are well written. My only suggestion would be that rather than saying ‘detection dogs’ your could say ‘detection dog teams’ or ‘detection dog-handler teams’ as the dogs do not work by themselves. Besides that suggestion, no modifications required.

-       -  In your introduction it would be good to highlight whether it is known if CWD poses any risk to the dogs themselves (especially if the faeces is eaten – which absolutely can happen in training if samples are not always housed in training equipment).

-        - In the introduction where you are discussing alternative training aids, it might be beneficial to refer to Needs et al. 2021 research (Do detection dogs respond differently to dried, frozen and live plant targets?) and if drying of the scats for training is an alternative training aid option.

-       -  In your methods you did not make reference to Table 1.

-        - I could not tell from your training protocols, but how did you account for the dogs not alerting to samples simply because they were novel? Were novel non-targets used often in training, such as those laboratory odours you mentioned in lines 181-182?

-       -  For your methods, it might be beneficial to include a figure that depicts the stages of training and then testing.

-        - Line 268: I believe this should be tables 2, 3 and 4.

-        - In your results section, beginning line 266, I would personally list what the actual sensitivity and specificity scores were (i.e. the percentages) in addition to what you have already provided. I think this would be beneficial in your tables as well or else it is hard to compare when certain variables had different scores (e.g. you had 25 CWD-negative samples with a 6 hour incubation time, but only 19 with a 48 hour incubation time).

-       -  In relation to lines 310-313: this still suggests to me that this training aids may not be capturing the complete odour profiles of the targets vs non-target, therefore making it harder for the dogs to discriminate. It would be very interesting to compare these samples using GCMS to the dog findings. I think it is important to highlight that there were quite high levels of false alerts, and not gloss over this just because the dogs also alerted to the CWD-positive samples.

-       -  Section beginning at line 325: I think this is excellent and very relevant to the real-world! Most handlers won’t simply wait for performance of a full trained alert, especially when variables have been changed.

-       -  I think your discussion is excellent. One thing I believe is missing is reference to how long each training aid can be used for before it needs to be disposed of and replaced. This will be another important factors for people to consider when comparing different training aid products.

Author Response

Thank you for your comments. Our responses are below.

-        - Simple summary and abstract both provide excellent information and are well written. My only suggestion would be that rather than saying ‘detection dogs’ your could say ‘detection dog teams’ or ‘detection dog-handler teams’ as the dogs do not work by themselves. Besides that suggestion, no modifications required.

Thank you, this change has been made.

In your introduction it would be good to highlight whether it is known if CWD poses any risk to the dogs themselves (especially if the faeces is eaten – which absolutely can happen in training if samples are not always housed in training equipment).

Information about this has been added to the introduction; dogs at at very low risk for contracted any prion disease, and coyotes have been shown to not contract it from eating infected meat. Together this suggests that CWD is a low risk to dogs.

In the introduction where you are discussing alternative training aids, it might be beneficial to refer to Needs et al. 2021 research (Do detection dogs respond differently to dried, frozen and live plant targets?) and if drying of the scats for training is an alternative training aid option.

It has been made clearer in the introduction that drying or freezing do not eliminate infectious prion presence. 

In your methods you did not make reference to Table 1.

A reference to Table 1 has been added in the Participants section. 

 I could not tell from your training protocols, but how did you account for the dogs not alerting to samples simply because they were novel? Were novel non-targets used often in training, such as those laboratory odours you mentioned in lines 181-182?

Every trial that contained a positive odor contained a novel positive aid, novel negative aid,  and an incubated control, all of which were novel to the dog. We also used novel fresh aids (unincubated) as non targets. We have made this clearer in the methods section and did some minor reorganization to make the training and testing process clearer. 

For your methods, it might be beneficial to include a figure that depicts the stages of training and then testing.

We have added a supplementary table that clarifies the stages of training and testing.

-        - Line 268: I believe this should be tables 2, 3 and 4.

Thank you, this has been changed. 

-        - In your results section, beginning line 266, I would personally list what the actual sensitivity and specificity scores were (i.e. the percentages) in addition to what you have already provided. I think this would be beneficial in your tables as well or else it is hard to compare when certain variables had different scores (e.g. you had 25 CWD-negative samples with a 6 hour incubation time, but only 19 with a 48 hour incubation time).

This makes sense, thank you, this has been done for each of the tables. I also changed the wording since I actually provided sensitivity and false alert data and did not provide sensitivity (where sensitivity percentages are just 1-false alert percentage). I think this makes the table cleaner to read since the percentage/proportion always refers to the same thing (alerts).

-       -  In relation to lines 310-313: this still suggests to me that this training aids may not be capturing the complete odour profiles of the targets vs non-target, therefore making it harder for the dogs to discriminate. It would be very interesting to compare these samples using GCMS to the dog findings. I think it is important to highlight that there were quite high levels of false alerts, and not gloss over this just because the dogs also alerted to the CWD-positive samples.

This is true, I have added in more information in section 3 under Dogs’ trained final alerts on samples comparing this study to our previous study sensitivity and false alert rates on fecal matter, suggesting that the false alert rates are higher here, potentially because the odor is not faithfully represented.

-       -  Section beginning at line 325: I think this is excellent and very relevant to the real-world! Most handlers won’t simply wait for performance of a full trained alert, especially when variables have been changed.

Thank you!

-       -  I think your discussion is excellent. One thing I believe is missing is reference to how long each training aid can be used for before it needs to be disposed of and replaced. This will be another important factors for people to consider when comparing different training aid products.

I have added this into the discussion starting at line 493. 

Reviewer 3 Report

Comments and Suggestions for Authors

ANIMALS REVIEW - Assessing different Chronic Wasting Disease Training Aids for Use With Detection Dogs

The study presented concern an interesting topic, poorly studied with standardized scientific protocols: the use of different training aids materials for the training of detection dogs on different disease related biological samples. The study is well conducted and the paper clear to read, however there are some revision I would consider necessary to have more detailed about the procedure adopted in the test and some minor revisions, see below.

Line 61: wrong comma between be and used.
Line 97: I would say inherent biases.
Line 98: what do you mean with “requirements for their trained final response”?

Line 98-100: I would suggest to include in the literature previous studies exploring the generalization of detection dogs from training targets to real samples.

Line 122: I would keep the below description without “Dog participant information”.

Line 148-151: The positive samples used to create the positive pool were selected based on dog performance in another study? You did not perform a chemical analysis that would confirm that those samples are actually positive?
Could the authors better argument with support of previous evidence why they chose to use pooled of positive and negative samples instead of singular samples?

Methods
Line 199-204: How many trials did the dog perform? How many trials per session?
It is not clear to me whether in each trial it was present only one kind of CWD training aids (es. GetXent) both positive and negative, please specify it better. In the test phase, dogs were never presented with the original training material but only with the never seen CWD training aids or controls, is it right? 

Line 206-207: For blank trials, it is stated above that the dogs were trained to sniff all the port and then go and sit on a platform. During the test, blank trials, were dogs rewarded to sniff all the ports and go to sit on the platform or just to sniff and pass every item in the line-up?

Line 222: What definition/ethogram was used by the coder to identify the behaviour “duration of time dogs spent sniffing each odor in the lineup”? Please specify.

Line 256-251: The explanation of the two models run is present above, I would suggest just reporting the results of the model that converged.

Line 266-268: Alerted incorrectly means that they performed a false positive alert (i.e. performing a trained alert response to the negative target)?

It would be useful to have a table or a scheme of how many trials dogs did with each training aid material and subsection for each training material with how many trials with each incubation time and temperature.

Lines 333-339: Why did you chose to consider the “change in behaviour” response as a yes/no (binomial) variable and not to take into account the duration of the behaviour as response variable?

The paragraph “change in behaviour analyses” would be more appropriate in the analyses section.

There are different spaces before and after the “=” in the report of the results (sometimes space, sometimes no space), please be consistent (es. Lines 353-359).

Line 393-395: Authors state that training aids could be use alongside CWD fecal matter for training CWD detection, however in this study the generalization from CWD fecal matter to training aids with CWD odor was tested and not the other way round: training detection dogs on training aids impregnated with CWD fecal matter and then test their performance on CWD fecal matter. I would suggest that authors highlight this missing step in the research about the topic.

Please check citations: Citations 4, 15, 19 missing informations

Author Response

Thank you for your comments. Our responses are below.

The study presented concern an interesting topic, poorly studied with standardized scientific protocols: the use of different training aids materials for the training of detection dogs on different disease related biological samples. The study is well conducted and the paper clear to read, however there are some revision I would consider necessary to have more detailed about the procedure adopted in the test and some minor revisions, see below.

Line 61: wrong comma between be and used.

Fixed

Line 97: I would say inherent biases.

Fixed.

Line 98: what do you mean with “requirements for their trained final response”?

This has been changed to “type of final response (e.g., sit, stand-and-stare) [28], and required duration of trained final response [29]” for clarification. 

Line 98-100: I would suggest to include in the literature previous studies exploring the generalization of detection dogs from training targets to real samples.

 This has been added to the introduction, thank you. 

Line 122: I would keep the below description without “Dog participant information”.

This has been edited.

Line 148-151: The positive samples used to create the positive pool were selected based on dog performance in another study? You did not perform a chemical analysis that would confirm that those samples are actually positive?

The samples were all tested using immunohistochemistry to confirm CWD diagnosis, as described in the sample procurement section. Of all the samples we had available, we chose to pool samples where dogs had high accuracy in detection. Those samples were previously confirmed positive using IHC. Clarifications have been made to the section fecal matter used to create training aids to clarify these points. 

Could the authors better argument with support of previous evidence why they chose to use pooled of positive and negative samples instead of singular samples?

Pooled samples were used to ensure that dogs’ performance on any given alternative aid was not dependent on the sample itself. Samples differ widely based on the deer age, sex, location (which influences diet) and disease progression. We know that dogs can smell the individual samples within the pooled samples, and using pooling allows us to standardize across the aids. Additionally, we could not have used a single positive or negative across all aids because there would not have been enough fecal material for eighteen different incubations, and incubations should not be done more than once with any given sample. 

Methods
Line 199-204: How many trials did the dog perform? How many trials per session?

This has been clarified in the participants section. 

The six dogs had all previously participated in scent detection studies at the Penn Vet Working Dog Center. All the dogs were all originally trained to detect a synthetic chemical training odor (Universal Detector Calibrant, or UDC) [30] on an eight-armed scent detection wheel. Dogs were trained to search the wheel in order. They performed a trained final response at the port containing the UDC odor (either a 2 second stand-and-stare, or a sit), as well as a ‘blank wheel’ behavior, which consisted of leaving the search area to sit on a table to indicate that the target odor was not present in the presented odor samples. The dogs’ correct behavior was marked with a conditioned secondary reinforcer (“clicker”) and dogs were rewarded with a primary reinforcer (food) for finding and correctly performing their trained final response behavior at the port containing the UDC odor. During these training sessions the dogs were trained to ignore laboratory odors, such as nitrile gloves, isopropyl alcohol, and paper towels.

Three of the privately owned dogs had previously been trained to find fecal samples from CWD-positive deer in laboratory and field settings. The additional three dogs were trained to differentiate fecal samples from CWD-positive and CWD-negative deer specifically for this alternative training aids study. All dogs achieved at least 80% sensitivity and 80% specificity on their initial response to odor across three practice sessions prior to participating in this study. The previously trained dogs required between 5 and 8 sessions of 8-10 trials each to achieve criteria. The three dogs that were trained specifically for this study took 24 sessions of 8-10 trials each to reach 80% sensitivity and 80% specificity across three sessions.

It is not clear to me whether in each trial it was present only one kind of CWD training aids (es. GetXent) both positive and negative, please specify it better. In the test phase, dogs were never presented with the original training material but only with the never seen CWD training aids or controls, is it right? 

This has been clarified in the Testing and Participant Training sections under Procedure. Thank you, it should be clearer now. 

Line 206-207: For blank trials, it is stated above that the dogs were trained to sniff all the port and then go and sit on a platform. During the test, blank trials, were dogs rewarded to sniff all the ports and go to sit on the platform or just to sniff and pass every item in the line-up?

This has been clarified - dogs were rewarded when they sat on the platform for blanks during test as well.

Line 222: What definition/ethogram was used by the coder to identify the behaviour “duration of time dogs spent sniffing each odor in the lineup”? Please specify.

As seen in the ethogram in Supplementary Table 3, duration at port is defined as “starts when dog’s nose is within 6 inches of port, ends when dog is further than 6 inches from port. If dog’s nose returns to port, duration is restarted”. As such, duration at port is the total amount of time in one trial that dogs spend with their nose closer than 6 inches to the port. 

Line 256-251: The explanation of the two models run is present above, I would suggest just reporting the results of the model that converged.

This model was a different model, so it is relevant to report that this different model did not converge either. 

Line 266-268: Alerted incorrectly means that they performed a false positive alert (i.e. performing a trained alert response to the negative target)?

Yes, alerted incorrectly means a false positive alert. We have made this clearer in the text. 

It would be useful to have a table or a scheme of how many trials dogs did with each training aid material and subsection for each training material with how many trials with each incubation time and temperature.

This has been included (Table 2); all dogs did one trial with each type of aid for a total of 18 trials. 

Lines 333-339: Why did you chose to consider the “change in behaviour” response as a yes/no (binomial) variable and not to take into account the duration of the behaviour as response variable?

We could not use duration as a response variable because we 1) marked and rewarded the dog if they alerted on the positive samples and 2) did not end the trial / correct the dog when they alerted on negative samples, thus removing our ability to use duration data. Our dogs were not trained on intermittent reward, meaning they had no experience searching and finding target items and not receiving a reward for alerting correctly. They also had to complete a great deal of trials, so this was the best method to complete the trials in a timely fashion with the dogs’ current training. If the dog is clicked at a sample at two seconds and rewarded, I then cannot compare it to a situation where the dog falses on a sample for 3 seconds. 

The paragraph “change in behaviour analyses” would be more appropriate in the analyses section.

This has been moved. 

There are different spaces before and after the “=” in the report of the results (sometimes space, sometimes no space), please be consistent (es. Lines 353-359).

This has been standardized.

Line 393-395: Authors state that training aids could be use alongside CWD fecal matter for training CWD detection, however in this study the generalization from CWD fecal matter to training aids with CWD odor was tested and not the other way round: training detection dogs on training aids impregnated with CWD fecal matter and then test their performance on CWD fecal matter. I would suggest that authors highlight this missing step in the research about the topic.

We have written a separate paper with the inverse (training on aids and testing on fecal matter) that we are planning to submit after this one; however, this is the more applicable version, as dogs are likely to be trained on fecal matter itself and the aids are meant to be used in situations where it would not be safe to use the fecal matter, rather than replacing the fecal matter entirely. Since the target material is fecal matter, it is important to keep as much fecal matter in training as possible.

Please check citations: Citations 4, 15, 19 missing informations

I have edited these citations, thank you.